# Future Role of Health Technology Assessment for Genomic Medicine in Oncology: A Canadian Laboratory Perspective

Don Husereau [1,*], Yvonne Bombard [2,3], Tracy Stockley [4,5], Michael Carter [6], Scott Davey [7,8,9], Diana Lemaire [10], Erik Nohr [11,12], Paul Park [13], Alan Spatz [14,15,16], Christine Williams [7,10], Aaron Pollett [17,18], Bryan Lo [19], Stephen Yip [20], Soufiane El Hallani [21] and Harriet Feilotter [7,10]

1. School of Epidemiology and Public Health, University of Ottawa, Ottawa, ON K1G 5Z3, Canada
2. Institute of Health Policy, Management and Evaluation, University of Toronto, Toronto, ON M5T 3M6, Canada; yvonne.bombard@utoronto.ca
3. Genomics Health Services Research Program, Li Ka Shing Knowledge Institute, Michael's Hospital, Unity Health Toronto, Toronto, ON M5B 1T8, Canada
4. Division of Clinical Laboratory Genetics, Laboratory Medicine Program, University Health Network, 200 Elizabeth Street, Toronto, ON M5G 2C4, Canada; tracy.stockley@uhn.ca
5. Department of Laboratory Medicine and Pathobiology, University of Toronto, 1 King's College Circle, Toronto, ON M5S 1A8, Canada
6. Department of Pathology and Laboratory Medicine, Nova Scotia Health (Central Zone), Halifax, NS B3H 1V8, Canada; michaeld.carter@nshealth.ca
7. Department of Pathology and Molecular Medicine, Queen's University, Kingston, ON K7L 3N6, Canada; scott.davey@queensu.ca (S.D.); christine.williams@oicr.on.ca (C.W.); hf4@queensu.ca (H.F.)
8. Division of Cancer Biology and Genetics, Queen's University Cancer Research Institute, Queen's University, Kingston, ON K7L 3N6, Canada
9. Departments of Oncology and Biomedical and Molecular Sciences, Queen's University Cancer Research Institute, Queen's University, Kingston, ON K7L 2V7, Canada
10. Ontario Institute for Cancer Research, 661 University Ave, Toronto, ON M5G 0A3, Canada
11. Department of Pathology and Laboratory Medicine, University of Calgary, Calgary, AB T2N 1N4, Canada; erik.nohr@albertaprecisionlabs.ca
12. Alberta Precision Laboratories, Foothills Medical Center, 1403 29 St NW, Calgary, AB T2N 2T9, Canada
13. Department of Pathology, Rady Faculty of Health Sciences, University of Manitoba, Winnipeg, MB R3A 1R9, Canada; ppark@sharedhealthmb.ca
14. Division of Pathology, McGill University Health Center, 1001 Decarie Blvd., Montreal, QC H4A 3J, Canada; alan.spatz@mcgill.ca
15. OPTILAB-MUHC & Department of Laboratory Medicine, 1001 Decarie Blvd., Montreal, QC H4A 3J, Canada
16. Research Molecular Pathology Center, Lady Davis Institute, 3755 Côte Ste-Catherine Road, Montreal, QC H3T 1E2, Canada
17. Pathology & Laboratory Medicine, Sinai Health System, Toronto, ON M5G 1X5, Canada; aaron.pollett@sinaihealth.ca
18. Laboratory Medicine & Pathobiology, University of Toronto, Toronto, ON M5S 1A8, Canada
19. The Ottawa General Hospital, 501 Smyth Road, Ottawa, ON K1H 8L6, Canada; brlo@toh.ca
20. Department of Pathology & Laboratory Medicine, Faculty of Medicine, University of British Columbia, Vancouver, BC V6T 1Z7, Canada; stephen.yip@vch.ca
21. Alberta Precision Laboratory, Department of Laboratory Medicine and Pathology, Faculty of Medicine and Dentistry, University of Alberta, Edmonton, AB T6G 2R7, Canada; elhallan@ualberta.ca
* Correspondence: dhuserea@uottawa.ca; Tel.: +1-6132994379

**Abstract:** Genome-based testing in oncology is a rapidly expanding area of health care that is the basis of the emerging area of precision medicine. The efficient and considered adoption of novel genomic medicine testing is hampered in Canada by the fragmented nature of health care oversight as well as by lack of clear and transparent processes to support rapid evaluation, assessment, and implementation of genomic tests. This article provides an overview of some key barriers and proposes approaches to addressing these challenges as a potential pathway to developing a national approach to genomic medicine in oncology.

**Keywords:** genomic biomarkers; clinical utility; clinical validity; analytic validity; health technology assessment

## 1. Introduction

Genome-based tests are increasingly becoming commonplace in healthcare. Their use is motivated by lower per-test costs and an emerging future of genomic medicine; this includes an increasing focus on biomarkers to aid in the screening, diagnosis, prognosis, and monitoring of disease; the use of targeted therapies; and the introduction of new testing modalities, including novel sampling (e.g., point-of-care) or analytic approaches (e.g., algorithms and multi-omic applications) [1,2].

Unlike most medical technology, which is associated with additional care expenditure, genome-based applications are associated with a wide range of cost and health impacts. Some have already proven to reduce healthcare costs while greatly improving care [3], while others represent significant costs with little health gain [4]. They may also have ethical and legal implications (e.g., privacy and autonomy) [2]. To incentivize the adoption and uptake of valuable innovation and improve decision making for technologies with broader societal implications, some jurisdictions in Canada and worldwide have applied existing health technology assessment (HTA) processes to focus on the applications of genomic medicine. However, approaches for evaluating diagnostic tests vary widely [5], and few have developed evaluative approaches specific to genome-based testing [6].

Nonetheless, international consensus recommendations for tailoring HTA processes to genome-based testing have been developed [6] and seek to address specific challenges in assessing genome-based testing [7,8], with an emphasis on HTA principles including transparency, stakeholder engagement, timeliness, and value-based pricing [9]. While this guidance is very helpful, it is also focused on evaluative considerations and falls short of addressing some larger HTA process and health system issues that surround complex interventions that may deeply impact patient care and health outcomes, including what timeframe for evaluation and adoption is appropriate, limitations with the use of published reports given the speed of technologic change and the need for real-world evidence, the role of education and training, and the need to account for care pathways that describe the availability and timing of human and capital resource requirements. Many of these gaps in traditional HTA frameworks, however, have been described in recommendations that address the larger context of HTA for medical devices [10].

HTA guidance for genome-based testing [6] also provides a telling suggestion for Canada and other federated (pluralistic) health systems with disjointed processes for decision making in HTA [11]. It suggests the development of "harmonized HTA requirements across national/regional HTA groups", where possible, "to enhance timely access to molecular diagnostics and streamline the process and reduce workload for manufacturers and HTA bodies" [6]. While not explicit in this guidance, harmonized HTA processes would also work toward reducing inequity of access to testing across healthcare regions, preventing a postal code lottery where patients benefit based on where they live [11].

### 1.1. Evaluation and Implementation of Genome-Based Testing in Canada

In Canada, this lottery already exists. Prioritization and funding of testing is highly regionalized by province, with important disparities (see Table 1). The largest provinces (by population, each >4.5 M), namely Ontario, Quebec, British Columbia, and Alberta, have each adopted one or more formal HTA processes to address genome-based testing that are at various stages of development. Smaller jurisdictions may rely on more informal processes such as clinical networks or hospital-based committees to aid decision making. They may also rely on the Canadian Agency for Drugs and Technologies in Health (CADTH) to provide information and advice to aid decision making. CADTH is funded by all provinces (except Quebec) and by Canada's Federal government with a mandate to provide "health

care decision-makers with objective evidence to help make informed decisions about the optimal use of health technologies, including ... diagnostic tests" [12]. In Quebec, the health-ministry-piloted Réseau Québécois de Diagnostic Moléculaire (RQDM) provides final recommendations to the health ministry for access and reimbursement of genomic testing in cancer and Mendelian genetics.

**Table 1.** Canadian organizations responsible for evaluation and implementation of testing, by province.

| Province | Size (M) | Health Technology Assessment Process (Organization(s) Responsible) | Program/Authority Responsible for Planning and Implementation |
|---|---|---|---|
| Alberta | 4.2 | Alberta Precision Laboratories Test Formulary Committee | Alberta Precision Laboratories/ Alberta Health Services |
| British Columbia | 5.0 | Provincial Laboratory Medicine Services Test Review Process | Provincial Health Services Authority |
| Manitoba | 1.3 | - | Winnipeg Regional Health Authority |
| New Brunswick | 0.8 | - | Horizon Health |
| Newfoundland and Labrador | 0.5 | - | Eastern Health |
| Nova Scotia | 1.0 | - | Nova Scotia Health Authority |
| Ontario | 14.2 | Provincial Genetics Advisory Committee/Ontario Genetic Advisory Committee/Program in Evidence-Based Care | Provincial Genetics Program/ Ontario Health |
| Prince Edward Island | 0.2 | - | Health PEI |
| Quebec | 8.5 | Institut national d'excellence en santé et en services sociaux | Direction de la biovigilance et de la biologie médicale (Ministry) |
| Saskatchewan | 1.1 | - | Saskatoon Health Region |

Even with a fit-for-purpose evaluative framework, challenges with implementation can introduce healthcare disparities within provinces, as provinces have typically relied on a mixture of different types of funding sources (e.g., research grants, donations to larger hospitals, and global hospital budgets) and service centres for implementation (e.g., highly specialized hospitals, and out-of-province contracting). This may result in a situation where a test is funded for all qualified patients within a province but its actual availability, the education surrounding its use, the time to return test results, or the analytic and proficiency standards surrounding it depend on where a person is treated [13–15].

Inconsistent (or nonexistent) processes to assess and approve genetic tests across Canada, coupled with the lack of consistent test standards, funding, and service delivery models, were already being described over a decade ago [16]. In the province of Ontario alone, calls for reform to address health system disparities have resulted in a number of audits [17,18], commissions [19,20], and multi-stakeholder discussions [21]. Recommendations to optimize testing in Ontario have been more recently published [15], and changes to its approach are underway, including the development of a test list and creation of a single service organization dedicated to genetic testing.

### 1.2. Evaluative Frameworks of Genome-Based Tests for Clinical Use

It appears to be broadly accepted that adoption of oncology biomarkers should be based on the parameters of clinical utility (i.e., What is the potential health or healthcare/economic impact of the biomarker test result?), clinical validity (To what extent does the biomarker track with the phenotype of interest?), and technical or analytic validity (Are we able to assess each biomarker consistently and accurately?). In practice, these tenets may be difficult to apply given the evolving evidence or the lack of direct evidence to support the adoption of many tests. Nonetheless, decisions must be made despite uncertainty—this, along with the use of different evaluative frameworks and processes, contributes to the messy landscape of oncology genomic biomarkers being used to assess Canadian patients.

While clinical utility is relatively simple to assess for predictive or companion biomarkers, this feature is much more difficult to assess for other types of biomarkers, including those that provide prognostic, diagnostic, or recurrence information, typically due to a lack of robust evidence to support claims. Funders across the country may have different approaches to assessing the overall utility of such biomarkers. Broadening the definition of clinical utility to include impacts on the system rather than only on the individual patient changes the landscape of how utility could be assessed in an evolving health care system. There may also be other benefits (e.g., access to community supports) that are not typically examined in HTA frameworks but are seen as valuable by clinicians [22], which results in a disconnect between benefits valued in policy vs. practice.

Clinical validity, defined here as the extent to which the clinical utility holds true for any biomarker, is generally not quantitatively assessed. It is likely an element that cannot be in place prior to the adoption of a biomarker into clinical use given that it is a constantly changing parameter depending on many other aspects of patient care. Clinical validity probably should be measured multiple times, i.e., prior to the adoption of a biomarker into clinical use and at intervals following the adoption, to ensure the marker continues to deliver value. The option to disinvest in approved biomarkers is a concept that relies on the ability to measure ongoing clinical validity, a task for which the Canadian health care system is not yet prepared.

Analytic or technical validity, perhaps the easiest parameter to comprehend, is often left to individual laboratories to define. Approval of a genomic biomarker for oncology should be predicated in part on the notion that labs are able to effectively and reproducibly measure the biomarker to a desired standard. In most Canadian provinces and territories, approval for funding of a new genomic oncology biomarker is accompanied by little more than instruction to identify mutations in Gene X. While standard metrics of analytic validity are clear, the actual details of the spectrum of clinically relevant mutations as well as the sensitivity/specificity parameters that would best guide clinical use of the test are often left to individual laboratories to determine. The not-unexpected consequence of this lack of guidance is a laboratory community offering tests whose metrics are based on individual laboratory resources and capabilities rather than according to provincial or national standards.

### 1.3. Health Technology Management through a Learning Health System (LHS)

A health system that takes a learning approach integrates clinical care with research so that real-time data are used to inform patient care and patient outcome data informs research. A learning health system could solve many of the issues around consistently delivering genome-based tests by providing the framework—provincially or nationally—based on the assumption that the clinical utility, clinical validity, and analytic validity as assessed will lead to significant uncertainty and precede decisions about funding or clinical adoption [20]. Further, ongoing assessment of the value of any biomarker being assessed for patients should be the key to an agile system that ceases testing for biomarkers that either do not offer the expected value or that have been replaced by newer biomarkers that do a better job [23]. Hand in hand with the need to disinvest in biomarkers that do not demonstrate clinical or cost effectiveness in the real world is the idea that the initial adoption of a biomarker should lean towards permissiveness. This would allow clinical adoption of more biomarkers with a clear expectation that funding for any given biomarker is dependent on its continued clinical utility. Measuring the clinical utility (and cost impacts), in turn, is a real-world evidence problem that is within the remit of provinces and could be the purview of a national organization such as CADTH, which has more recently launched initiatives in real-world evidence [24].

While a proposal for both a common test-review process [16] and a life-cycle health technology assessment (i.e., health technology management) approach using real-world data and a learning healthcare system has been described (for "precision oncology", more specifically) [25], there is little attention to more specific details regarding health technology

management, including how tests are considered, triaged, and implemented. Borrowing from an already-described framework for good practices in HTA [26], we highlight some key features that should be considered in the Canadian context along with key considerations (Table 2).

**Table 2.** Key features of a Canadian health technology process for genome-based tests in oncology.

| Goal | Good Practices in HTA, from [26] | Subdomain | Proposed Feature(s) | Rationale |
|---|---|---|---|---|
| Adequate level of support for decision making by asking appropriate questions and providing timely recommendations | Defining the HTA process | Structure and governance | Process developed through multi-stakeholder deliberation | To enhance perceived legitimacy of the process |
| | | | Pan-Canadian/linked to provinces and provincial lab programs | Promoting consistent decision making across provinces |
| | | Framing and scoping | Pre-defined scoping that considers use of biomarker | Different biomarker uses require different evaluative approaches |
| | | | Horizon scanning and a single point of entry | Testing requires ample lead-time for implementation |
| | | | Rapid priority setting | To ensure timely access to life-saving tests or tests with urgent patient needs |
| Provide consistency in evaluating costs and benefits of testing | Assessment | | Rapid review and real-world assessment | To ensure timely access to life-saving tests or tests with urgent patient needs |
| Provide high-value and equitable testing | Contextualization | | Province-led contextualization of implementation | To ensure testing and its potential value is fit for purpose |
| Reduce duplication, improve timeliness, and adapt to changing innovation. | Implementation and monitoring | | Centralized evaluation of the measurement process | To remove burden on individual clinical labs to fund optimization studies |
| | | | Generate standardized life-cycle evidence | To improve value while adapting to constantly changing landscape of innovation |

*1.4. Key Features of a Pan-Canadian Health Technology Assessment Process*

1.4.1. Overall Process Developed through Multi-Stakeholder Deliberation

Beyond the payer and laboratory professional community, the adoption and use of new testing will have impacts on multiple stakeholders, including patients, care providers, private and public innovators, researchers, and others. Consistent with good practices [27], any overarching approach to life-cycle health technology assessment requires an exchange of perspectives among key stakeholders.

1.4.2. Pan-Canadian/Linked to Provinces and Provincial Lab Programs

In a similar vein, HTA is not useful if it is not directly linked to decision makers; key to an effective system of pan-Canadian HTA is buy-in from local decision-making authorities, including newly emerging provincial programs dedicated to testing or genetic testing.

1.4.3. Pre-Defined Scoping That Considers the Use of Biomarker Tests

An agile provincial or pan-Canadian framework will require creating standardized questions about different types of genomic biomarker tests (i.e., predictive, prognostic, recurrence, and diagnostic) ahead of time as a means to aiding rapid assessment. This could be implemented via an oversight committee, which could also aid in triaging proposals for new tests into those requiring standard HTA assessment, versus those that clearly meet needs, to avoid delays to timely access.

1.4.4. Horizon Scanning and a Single Point of Entry to Consider New Biomarkers

New genomic biomarkers are developed by researchers in the academic or private sector. An ongoing frustration of these developers is the lack of a clear entry point into the

health care system. This could be solved by having a clear and single point of entry into the health care system that mandates an evaluation (as opposed to prioritization, where a proposal can be shelved). A similar approach can be seen with drug review, where applications for new technology proposals are open to payers, clinicians, and private sector innovators. Researchers, from both the academic and private sectors, could also provide early indicators that a new biomarker is being developed through a systematic program of horizon scanning [28]. As well as providing clarity for researchers developing these biomarkers, this approach would better allow labs and health system funders to see what is coming and better manage the continued adoption of expensive lab assays by a more anticipatory approach to adoption instead of using a reactive, one-test-at-a-time approach. This, in turn, would reduce the strain on issues around capital infrastructure and human resource planning.

### 1.4.5. Rapid Priority Setting and Triage

Within the research space, development of biomarkers for clinical use requires testing, validation, and preclinical studies. Unlike drug development, however, many genomic biomarkers are not associated with randomized clinical trial data. Therefore, the type of evaluative evidence available to assess a biomarker is unlikely to be as robust as the level of evidence used for drug assessment. Evaluative frameworks that include agreed-upon standards and tools (such as a checklist) for evaluation of new genomic biomarkers would be a resource that could be used by any or all health care authorities in Canada. Agreed-upon tools like this can be the first steps towards the standardization of testing and a better understanding of what level of evidence will be required by funders. Further development of consistent standards for evaluation will also need to recognize that different types of biomarkers will require different types of evidence. Predictive, prognostic, risk, screening, diagnostic, pharmacogenomic, early detection, or residual disease biomarkers will each require a specific template or approach. Moreover, some biomarkers may require higher levels of urgency for patients and a more rapid evaluation by payers.

### 1.4.6. Rapid Review and Real-World Assessment

Inherent in the idea of a checklist that could be used by the funder (or the HTA body supporting them) to evaluate the readiness of a genetic test for clinical use is that the checklist itself would be sufficient to support decision making, avoiding the traditional health technology assessment bottleneck. While this could be true for many genetic biomarkers, in particular those associated with drug response, some biomarkers may not be readily evaluated using these same metrics. An oversight committee who could make rapid decisions about the need to invoke more traditional HTA processes would allow continued support of established and well-functioning HTA groups while not burdening them with assessment of tests that do not require that type of scrutiny. This would have the advantage of standardizing the approach for adoption of any biomarker or test while retaining the ability to use HTA resources more wisely.

### 1.4.7. Province-Led Contextualization

As with the current pan-Canadian reimbursement review process for drugs, individual provincial priorities across a federated health system would be expected to vary. While a pan-Canadian approach to rapid triage and implementation is ideal for promoting consistency in evidence-based policy, individual provinces would likely still need to decide which tests are a priority for implementation given current healthcare system demands. This is not to suggest tests will be available according to postal code/province. To mitigate concerns regarding equity of access to testing, provinces could still decide to reimburse testing through out-of-province arrangements (as they do now) prior to implementation.

### 1.4.8. Centralized Evaluation of the Measurement Process

Traditionally, in Canada, biomarkers that are approved for use and funding are still missing a critical evaluation, which is specific recommendations regarding their analytic validity and performance standards. The ability to effectively and reproducibly measure a biomarker before it is funded in multiple clinical labs is a key feature of assessment that can be overlooked. That is, while HTA may look at what performance has been achieved (i.e., descriptively), this is not as useful as recommending what performance is required (i.e., normatively). And while assessments may imply minimum levels of performance, decision makers may in turn provide laboratories with little more than the names of genes or proteins that require study, with no guidance as to which aspects of those molecules are important and at what level of sensitivity or specificity the clinical value lies.

Part of the assessment of the suitability of a biomarker for use must include evaluation of the measurement process. Ideally, this could be done in a centralized or virtual laboratory network that could work for the country. Such a lab could evaluate multiple commercial or lab-developed solutions for measuring a biomarker under consideration, produce information to harmonize results from multiple assays, and develop standardized quality metrics for a new biomarker. This information could be shared with provincial and territorial funders and then, if approved, with clinical labs interested in offering testing. It is important to note that this implementation phase does not remove the need for each lab to carry out a clinical validation of their assay of choice, but it could go a long way to removing the burden on individual clinical labs to fund expensive and exhaustive studies on the best way to measure a new biomarker.

### 1.4.9. Generate Standardized Life-Cycle Evidence

A health care system that is permissive (i.e., adopts new biomarkers or tests with a minimum of high-quality evidence) will adopt some tests that, while promising through investigation, do not achieve the desired clinical impact in practice. Other tests will become outdated or surpassed by new markers or approaches. A learning health system must be able to divest itself of such low-performance tests at least as rapidly as it brings on new tests. To do this requires effective real-world studies that make use of granular and standardized data to evaluate lab test effectiveness. While CADTH provides real-world evidence expertise, the fact is that even knowing which data elements to collect and where to obtain them remains an obstacle. To begin to unravel this problem, each biomarker or test that is approved for study should come with a list of specific data elements that should be collected and collated for real-world studies. Other details regarding this process have been further described elsewhere [25].

### 1.4.10. Meeting End-User and Societal Needs

Ultimately, the measure of success lies in how well the system created meets the needs of end users, namely patients and clinicians and the public that funds its implementation. Patients need to be well informed to play an active role in their health care decisions, while clinicians rely on health technology to answer questions related to patient management. In a patient-centred care environment, clinicians and patients plan the treatment journey together using relevant and accessible information to make decisions that are right for the patient, ranging from standard-of-care through to experimental therapeutic interventions.

The assessment of the value of biomarker testing must take into account patient values, preferences, and needs. While biomarkers may provide information about treatment options or diagnosis, for the patient, the value of a test's information may be more complex [29] and may include the impact on patient psychology, including whether the result dispels anxiety or provokes it when no effective treatment is available [30], or even the misuse or misunderstanding of the power of results [8,31].

Patients also value the sense of empowerment that comes with understanding the meaning of testing that has been performed. This can be facilitated by reporting that is accessible and clear to patients [32]. Patient education around biomarker testing and

the implications of results as well as easy and real-time access to those results [33] are essential. Results can give a patient the confidence that everything that could be done is being done, and understanding the results enhances that perception. Therefore, any solution for bringing biomarkers to clinical use must take into account how patients will be engaged in the process, from easy access to their test results and from access to effective educational materials [34–36].

Certainly, many issues that are important to patients will surface when patient engagement is carried out appropriately and consistently so that patients can incorporate their values and needs into the assessment process, and these continue to be reflected following implementation. Recently, CADTH has introduced a values-based framework for patient engagement in health technology assessment to involve patients, families, and patient groups in improving the quality and relevance of assessments.

There are additionally a number of needs that may not directly affect patients but have larger ethical and legal implications for others. These include considerations of autonomy, privacy, confidentiality, and equity, which can impact family members and broader society. Many of these issues have been previously described [7,37].

## 2. Concluding Remarks

While a standardized pan-Canadian approach to the evaluation and implementation of testing is desirable, given the increasing demand for genome-based testing and the need for equitable and timely access to tests, there is still considerable work to be done to achieve this goal. The current patchwork of evaluative processes and oversight for testing is still in development. Harmonizing these may require further maturation of these processes so that individual organizations see the benefits of collaborating, or they could occur through larger, pan-Canadian incentives.

While we have not outlined what has to happen for provinces to work together, it is clear there are considerable benefits in doing so. The development of a fit-for-purpose evaluative system will lend itself to more equitable opportunities for improving population health, better patient and care provider experiences, and improving health system efficiency for all Canadians regardless of where they live.

**Author Contributions:** Conceptualization, D.H. and H.F.; writing—original draft preparation, D.H. and H.F.; writing—review and editing, D.H., Y.B., T.S., M.C., S.D., D.L., E.N., P.P., A.S., C.W., A.P., B.L., S.Y., S.E.H. and H.F. All authors have read and agreed to the published version of the manuscript.

**Funding:** This research was funded by the Canadian Cancer Society through an Accelerator grant held by H.F. (Grant 707637 awarded in 2022).

**Institutional Review Board Statement:** Not applicable.

**Informed Consent Statement:** Not applicable.

**Data Availability Statement:** No new data were created or analyzed in this study. Data sharing is not applicable to this article.

**Acknowledgments:** Y.B. holds the Canada Research Chair in Genomics Health Services and Policy. D.L. is a patient expert and her authorship is intended to ensure patient experiences, priorities and preferences are appropriately represented in this work.

**Conflicts of Interest:** The authors declare the following conflict of interest. D.H. has received funding, outside the submitted work, from pharmaceutical and diagnostic companies with an interest in genome-based testing and precision medicine, including Amgen Canada Inc., AstraZeneca Canada, Eli Lilly Canada Inc., GlaxoSmithKline Inc. (GSK Canada), Janssen Inc./J&J, Pfizer Canada ULC, Thermo Fisher Scientific Inc., and Roche Canada. Y.B. is cofounder of Genetics Adviser. T.S. has been a consultant for Health Canada (current) Advisory boards (with honoraria) within the past 36 months, related to diagnostic genetics, from AstraZeneca, Janssen, Bayer, Pfizer, and Merck. M.C. has participated in advisory boards and/or received speaking honoraria from Amgen, Incyte, Janssen, Merck, Novartis, and Pfizer. E.N. and S.E.H. both receive funding from Alberta Precision Laboratories E.N. has received an honorarium from Bayer Pharmaceuticals and Grants/Research Support from

AstraZeneca. D.L., P.P., A.S., C.W., A.P., S.Y., S.E.H., and H.F. have no conflicts to disclose that are relevant to this article.

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
