# Peer review of "Future Role of Health Technology Assessment for Genomic Medicine in Oncology: A Canadian Laboratory Perspective"

_curroncol, doi:10.3390/curroncol30110700_

Round 1
Reviewer 1 Report
Comments and Suggestions for Authors
Review report
Title:
Future role of Health Technology Assessment for Genomic Medicine in Oncology: A Canadian Laboratory Perspective
Authors:
Don Husereau , Yvonne Bombard , Tracy Stockley , Michael Carter , Scott Davey , Diana Lemaire , Erik Nohr , Paul Park , Alan Spatz , Christine Williams , Aaron Pollett , Bryan Lo , Stephen Yip , Soufiane El Hallani , Harriet Feilotter
Summary:
The article summarizes key barriers of the efficient and considered adoption of novel genomic medicine testing in Canada. It describes a potential health technology management strategy, the Learning Health System. The article gives an overview of the evaluation and implementation of genome-based testing in Canada. It outlines a standardized pan-Canadian approach to the evaluation and implementation of genome-based testing, by describing the Pan-Canadian Health Technology Assessment Process. The goal of the article is the enhancement of clinical implementation of genomic testing in Canada, despite healthcare and funding disparities within the provinces.
General concept comments:
The manuscript is scientifically sound, and the design is clear. The manuscript is relevant to the field and presented in a well-structured manner. References are pertinent, the cited references are recent publications and relevant. The manuscript does not include an excessive number of self-citations. The tables are relevant, they correctly show the data however, need minor improvement. Data is interpreted clearly in most cases and consistently throughout the manuscript. Conclusions are consistent with the evidence and arguments presented.
Specific comments and suggestions for Authors:
186. Table 2. Visualization of the content could be improved, the text is too close.
Comments on the Quality of English Language
102.
Table 1.
Heading: Health technology assessment process? (organization(s) responsible)
– Do we need the question mark?
189. No “dot” needed at the end of the subtitle
26. “asseessments” – Please correct.
Author Response
Specific comments and suggestions for Authors:
- Table 2. Visualization of the content could be improved, the text is too close.
RESPONSE: We have reformatted the table. Flagging for editors as well as this is based on the MDPI template
Comments on the Quality of English Language
102.
Table 1.
Heading: Health technology assessment process? (organization(s) responsible)
– Do we need the question mark?
RESPONSE: Have removed the “?”
- No “dot” needed at the end of the subtitle
RESPONSE: Have removed the “.”
- “asseessments” – Please correct.
RESPONSE: Have corrected
Reviewer 2 Report
Comments and Suggestions for Authors
The paper by Husereau and colleagues titled “Future role of health technology assessment for genomic medicine in oncology: the Canadian laboratory perspective“ provides a description of the growing need for genomic testing and its potential impacts and why an HTA process specifically tailored for genomic-based testing is required. The paper describes the current “messy landscape” of genomic testing in Canada and why a harmonized pan-Canadian approach is needed to avoid access to testing by postal code lottery. The paper also describes the wide range of funding resources that have to be cobbled together by institution/provinces to enable genomic testing to occur.
This background provides the rationale for the authors to argue for a health technology process that has a number of key features which include developing the overall process through a multi-stakeholder deliberation; a pan-Canadian approach linked to the provinces and their provincial laboratory programs; predefined scoping that considers the use of the biomarker tests to aid in their rapid assessment; a single point of entry to consider new biomarkers and a horizon scanning capability; rapid priority setting and triage; the opportunity for provinces to individually set priorities for biomarker test adoption based on health system demands; a centralized process to evaluate the optimal way of performing the tests; and a life cycle review of evidence for the utility of the biomarker tests based on effective real-world studies.
Table 1 shows what each of the Canadian provinces has in place for the evaluation and implementation of testing. For the majority of provinces, there is currently nothing in place and in the case of the largest province by population, Ontario, the province is at a very early stage of organizing for genomic testing. So the recommendations in this paper are timely and valuable and may help to move decision-making for a pan- Canadian HTA process for genomic tests forward. For several decades now, there has been no clear process for adopting and funding new biomarker tests that are increasingly needed for the selection of appropriate therapies for cancer patients. This has been incredibly frustrating for innovators, clinicians, patients, cancer agencies and individual institutions.
Although most of the recommendations for the HTA process are well thought through, this reviewer does not believe that province-led contextualization is something that should be encouraged. Rather, the opposite should be the tact, unless we are prepared to accept postal code implementation across the country. This reviewer would suggest leaving this out of the health technology process for genome -based testing in the text and in Table 2.
The paragraph on clinical utility (line 140 – 148) might be made clearer by an explicit example of how the clinical validity of a biomarker is not generally quantitatively assessed.
The paragraph on health technology management through learning health system appears somewhat contradictory as it as it emphasizes the importance of ongoing assessment of the value of any biomarker but suggests that the framework at the provincial and national level should assess the clinical utility, clinical validity and analytic validity before any funding or clinical adoption. It would seem more reasonable to this reviewer that a learning environment would put in place a conditional approval and funding mechanism with reassessment based on subsequent learnings and reevaluation of the clinical utility etc. this process might parallel CADTH’s recently introduced Procedures for Time-limited Reimbursement Recommendations.
Comments on the Quality of English LanguageA few minor comments:
Please explain “future proofing” in line 219 of the document.
Table 2 should be formatted in such a way that words are not broken apart making it hard to read. Perhaps using a landscape format would avoid this issue.
Somewhat greater spacing between the description of the health technology assessment processes in Alberta and British Columbia in Table 1 would be helpful.
“Responsible” is misspelled in the title to Table 1
The sentence beginning on 203 which repeats “oversight” twice might be stated differently.
Line 229 introduces the first person plural into the text which is not consistent with how the rest of the document was written.
Author Response
The paper by Husereau and colleagues titled “Future role of health technology assessment for genomic medicine in oncology: the Canadian laboratory perspective“ provides a description of the growing need for genomic testing and its potential impacts and why an HTA process specifically tailored for genomic-based testing is required. The paper describes the current “messy landscape” of genomic testing in Canada and why a harmonized pan-Canadian approach is needed to avoid access to testing by postal code lottery. The paper also describes the wide range of funding resources that have to be cobbled together by institution/provinces to enable genomic testing to occur.
This background provides the rationale for the authors to argue for a health technology process that has a number of key features which include developing the overall process through a multi-stakeholder deliberation; a pan-Canadian approach linked to the provinces and their provincial laboratory programs; predefined scoping that considers the use of the biomarker tests to aid in their rapid assessment; a single point of entry to consider new biomarkers and a horizon scanning capability; rapid priority setting and triage; the opportunity for provinces to individually set priorities for biomarker test adoption based on health system demands; a centralized process to evaluate the optimal way of performing the tests; and a life cycle review of evidence for the utility of the biomarker tests based on effective real-world studies.
Table 1 shows what each of the Canadian provinces has in place for the evaluation and implementation of testing. For the majority of provinces, there is currently nothing in place and in the case of the largest province by population, Ontario, the province is at a very early stage of organizing for genomic testing. So the recommendations in this paper are timely and valuable and may help to move decision-making for a pan- Canadian HTA process for genomic tests forward. For several decades now, there has been no clear process for adopting and funding new biomarker tests that are increasingly needed for the selection of appropriate therapies for cancer patients. This has been incredibly frustrating for innovators, clinicians, patients, cancer agencies and individual institutions.
Although most of the recommendations for the HTA process are well thought through, this reviewer does not believe that province-led contextualization is something that should be encouraged. Rather, the opposite should be the tact, unless we are prepared to accept postal code implementation across the country. This reviewer would suggest leaving this out of the health technology process for genome -based testing in the text and in Table 2.
RESPONSE: thanks for the comment which is warranted. Instead of removing the text, we have modified it. We are not suggesting as with drugs that reimbursement of testing be staggered across provinces. We had assumed provinces without capacity or priority to implement a test would use out-of-province billing arrangements as they do now. The following text was added
While a pan-Canadian approach to rapid triage and implementation is ideal for promoting consistency in evidence-based policy, individual provinces would likely still need to decide which tests are a priority for implementation given current healthcare system demands. This is not to suggest tests will be available according to postal code/province. To mitigate concerns regarding equity of access to testing, provinces could still decide to reimburse testing through out-of-province arrangements (as they do now) prior to implementation.
The paragraph on clinical utility (line 140 – 148) might be made clearer by an explicit example of how the clinical validity of a biomarker is not generally quantitatively assessed.
RESPONSE: TBD- Does someone want to offer an example?
The paragraph on health technology management through learning health system appears somewhat contradictory as it as it emphasizes the importance of ongoing assessment of the value of any biomarker but suggests that the framework at the
provincial and national level should assess the clinical utility, clinical validity and analytic validity before any funding or clinical adoption. It would seem more reasonable to this reviewer that a learning environment would put in place a conditional approval and funding mechanism with reassessment based on subsequent learnings and reevaluation of the clinical utility etc. this process might parallel CADTH’s recently introduced Procedures for Time-limited Reimbursement Recommendations.
RESPONSE: Agreed. The intent was never to suggest the assessment of biomarkers (utility , validity etc) must result in complete information, but rather that it would capture how much is unknown. This could then lead to RWE ot other mechansisms to resolve uncertainty. Some text has been added to qualify this in the paragraph:
“A health system that takes a learning approach integrates clinical care with research so that real-time data are used to inform patient care, and patient outcome data informs research. A learning health system could solve many of the issues around consistently delivering genome-based tests by providing the framework- provincially or nationally- based on the assumption that the clinical utility, clinical validity and analytic validity as assessed will lead to significant uncertainty and precede decisions about funding or clinical adoption [20]
A few minor comments:
Please explain “future proofing” in line 219 of the document.
RESPONSE: We have changed the text top avoid the jargon.
As well as providing clarity for researchers developing these biomarkers, this approach would better allow labs and health system funders to see what is coming, and better manage the continued adoption of expensive lab assays by a more anticipatory approach to adoption, instead of using a reactive, one-test-at-a-time approach. This, in turn, would reduce the strain on issues around capital infrastructure and human resource planning.
Table 2 should be formatted in such a way that words are not broken apart making it hard to read. Perhaps using a landscape format would avoid this issue.
Somewhat greater spacing between the description of the health technology assessment processes in Alberta and British Columbia in Table 1 would be helpful.
RESPONSE: We have made some formatting changes and will flag for editors.
“Responsible” is misspelled in the title to Table 1
RESPONSE: The spelling appears to be correct now.
The sentence beginning on 203 which repeats “oversight” twice might be stated differently.
RESPONSE: We have corrected.
Line 229 introduces the first person plural into the text which is not consistent with how the rest of the document was written.
RESPONSE: We have corrected.
Reviewer 3 Report
Comments and Suggestions for Authors
The manuscript by Husereau et al. is a clearly written, important paper describing the present state of, challenges in and future improvements for procedures in Health Technology Assessment (HTA) for genomic medicine in oncology in Canada. The topic is very timely as genomic medicine is being implemented with a high speed not only in countries with ample resources, but also in countries with moderate or low income, where the necessary organizational and operational regulatory requirements may be less developed. The critical overview of a real world Canadian experience complementing the International consensus recommendations for HTA processes in genome-based testing, therefore, is of great relevance to the international readership. The paper is also comprehensive addressing organizational, clinical and laboratory aspects of evaluation and implementation of processes, the requirements for an effective and timely integration of clinical care with research by Health Technology Management through a Learning Health System, and detailed features, short-comings and improvable targets of Pan-Canadian HTA processes.
A minor recommendation (optional): Since the manuscript apparently is not only based on literature survey, but also on some personal investigation and experience, this reviewer would recommend the insertion of a short methodological section on what preparative works were done and how the conclusions were drawn for the paper.
Author Response
The manuscript by Husereau et al. is a clearly written, important paper describing the present state of, challenges in and future improvements for procedures in Health Technology Assessment (HTA) for genomic medicine in oncology in Canada. The topic is very timely as genomic medicine is being implemented with a high speed not only in countries with ample resources, but also in countries with moderate or low income, where the necessary organizational and operational regulatory requirements may be less developed. The critical overview of a real world Canadian experience complementing the International consensus recommendations for HTA processes in genome-based testing, therefore, is of great relevance to the international readership. The paper is also comprehensive addressing organizational, clinical and laboratory aspects of evaluation and implementation of processes, the requirements for an effective and timely integration of clinical care with research by Health Technology Management through a Learning Health System, and detailed features, short-comings and improvable targets of Pan-Canadian HTA processes.
A minor recommendation (optional): Since the manuscript apparently is not only based on literature survey, but also on some personal investigation and experience, this reviewer would recommend the insertion of a short methodological section on what preparative works were done and how the conclusions were drawn for the paper.
RESPONSE: No change. But appreciate the suggestion.
Round 2
Reviewer 2 Report
Comments and Suggestions for Authors
My previously identified concerns have been addressed satisfactorily